# A Scale Free Algorithm for Stochastic Bandits with Bounded Kurtosis

**Tor Lattimore**[*]
tor.lattimore@gmail.com

## Abstract

Existing strategies for finite-armed stochastic bandits mostly depend on a parameter of scale that must be known in advance. Sometimes this is in the form of a bound on the payoffs, or the knowledge of a variance or subgaussian parameter. The notable exceptions are the analysis of Gaussian bandits with unknown mean and variance by Cowan et al. [2015] and of uniform distributions with unknown support [Cowan and Katehakis, 2015]. The results derived in these specialised cases are generalised here to the non-parametric setup, where the learner knows only a bound on the kurtosis of the noise, which is a scale free measure of the extremity of outliers.

## 1 Introduction

SpaceBandits is a fictional company that specialises in optimising the power output of satellite-mounted solar panels. The data science team wants to use a bandit algorithm to adjust the knobs on a legacy satellite, but they don't remember the units of the sensors, and have limited knowledge about the noise distribution of the panel output or sensors. The SpaceBandits data science team searches the literature for an algorithm that does not depend on the scale or location of the means of the arms, and find this simple paper, in NIPS 2017.

It turns out that logarithmic regret is possible for finite-armed bandits with no assumptions on the noise of the payoffs except for a known finite bound on the kurtosis, which corresponds to knowing the likelihood/magnitude of outliers [DeCarlo, 1997]. Importantly, the kurtosis is independent of the location of the mean and scale of the central tendency (the variance). This generalises the ideas of Cowan et al. [2015] beyond the Gaussian case with unknown mean and variance to the non-parametric setting.

The setup is as follows. Let $k \geq 2$ be the number of bandits (or arms). In each round $1 \leq t \leq n$ the player should choose an action $A_t \in \{1, \ldots, k\}$ and subsequently receives a reward $X_t \sim \nu_{A_t}$, where $\nu_1, \ldots, \nu_k$ are a set of distributions that are not known in advance. Let $\mu_i$ be the mean payoff of the $i$th arm and $\mu^* = \max_i \mu_i$ and $\Delta_i = \mu^* - \mu_i$. The regret measures the expected deficit of the player relative to the optimal choice of distribution:

$$\mathcal{R}_n = \mathbf{E}\left[\sum_{t=1}^{n} \Delta_{A_t}\right] . \tag{1}$$

The table below summarises many of the known results on the optimal achievable asymptotic regret under different assumptions on $(\nu_i)_i$. A reference for each of the upper bounds is given in Table 1, while the lower bounds are mostly due to Lai and Robbins [1985] and Burnetas and Katehakis [1996]. An omission from the table is when the distributions are known to lie in a single-parameter exponential family (which does not fit well with the columns). Details are by Cappé et al. [2013].

---

[*]Now at DeepMind, London.

| | Assumption | Known | Unknown | $\lim_{n\to\infty} \mathcal{R}_n / \log(n)$ |
|---|---|---|---|---|
| 1 | Bernoulli<br>Lai and Robbins [1985] | $\mathrm{Supp}(\nu_i) \subseteq \{0,1\}$ | $\mu_i \in [0,1]$ | $\sum_{i:\Delta_i>0} \dfrac{1}{d(\mu_i, \mu^*)}$ |
| 2 | Bounded<br>Honda and Takemura [2010] | $\mathrm{Supp}(\nu_i) \subseteq [0,1]$ | distribution | it's complicated |
| 3 | Discrete<br>Burnetas and Katehakis [1996] | $\mathrm{Supp}(\nu_i) \subseteq A$<br>$|A| < \infty$ | distribution | it's complicated |
| 4 | Semi-bounded<br>Honda and Takemura [2015] | $\mathrm{Supp}(\nu_i) \subseteq (-\infty, 1]$ | distribution | it's complicated |
| 5 | Gaussian (known var.)<br>Katehakis and Robbins [1995] | $\nu_i = \mathcal{N}(\mu_i, \sigma_i^2)$ | $\mu_i \in \mathbf{R}$ | $\sum_{i:\Delta_i>0} \dfrac{2\sigma_i^2}{\Delta_i}$ |
| 6 | Uniform<br>Cowan and Katehakis [2015] | $\nu_i = \mathcal{U}(a_i, b_i)$ | $a_i, b_i$ | $\sum_{i:\Delta_i>0} \dfrac{\Delta_i}{\log\left(1 + \frac{2\Delta_i}{b_i - a_i}\right)}$ |
| 7 | Subgaussian<br>Bubeck and Cesa-Bianchi [2012] | $\log M_{\nu_i}(\lambda) \le \frac{\lambda^2 \sigma_i^2}{2} \; \forall \lambda$ | distribution | $\sum_{i:\Delta_i>0} \dfrac{2\sigma_i^2}{\Delta_i}$ |
| 8 | Known variance<br>Bubeck et al. [2013] | $\mathbf{V}[\nu_i] \le \sigma_i^2$ | distribution | $O\left(\sum_{i:\Delta_i>0} \dfrac{\sigma_i^2}{\Delta_i}\right)$ |
| 9 | Gaussian<br>Cowan et al. [2015] | $\nu_i = \mathcal{N}(\mu_i, \sigma^2)$ | $\mu_i \in \mathbf{R}, \sigma_i^2 > 0$ | $\sum_{i:\Delta_i>0} \dfrac{2\Delta_i}{\log\left(1 + \Delta_i^2/\sigma_i^2\right)}$ |

$d(p,q) = p\log(p/q) + (1-p)\log((1-p)/(1-q))$ and $M_\nu(\lambda) = \mathbf{E}_{X\sim\nu} \exp((X-\mu)\lambda)$ with $\mu$ the mean of $\nu$ is the centered moment generating function. All asymptotic results are optimal except for the grey cells.

**Table 1:** Typical distributional assumptions and asymptotic regret

With the exception of rows 6 and 9 in Table 1, all entries essentially depend on some kind of scale parameter. Missing is an entry for a non-parametric assumption that is scale free. This paper fills that gap with the following assumption and regret guarantee.

**Assumption 1.** There exists a known $\kappa_\circ \in \mathbf{R}$ such that for all $1 \le i \le k$, the kurtosis of $X \sim \nu_i$ is at most $\mathrm{Kurt}[X] = \mathbf{E}[(X - \mathbf{E}[X])^4]/\mathbf{V}[X]^2 \le \kappa_\circ$.

**Theorem 2.** *If Assumption 1 holds, then the algorithm described in §2 satisfies*

$$\limsup_{n\to\infty} \frac{\mathcal{R}_n}{\log(n)} \le C \sum_{i:\Delta_i>0} \Delta_i \left(\kappa_\circ - 1 + \frac{\sigma_i^2}{\Delta_i^2}\right),$$

*where $\sigma_i^2$ is the variance of $\nu_i$ and $C > 0$ is a universal constant.*

What are the implications of this result? The first point is that the algorithm in §2 is scale and translation invariant in the sense that its behaviour does not change if the payoffs are multiplied by a positive constant or shifted. The regret also depends appropriately on the scale so that multiplying the rewards of all arms by a positive constant factor also multiplies the regret by this factor. As far as I know, this is the first scale free bandit algorithm for a non-parametric class. The assumption on the boundedness of the kurtosis is much less restrictive than assuming an exact Gaussian model (which has kurtosis 3) or uniform (kurtosis 9/5). See Table 2 for other examples.

As mentioned, the kurtosis is a measure of the likelihood/existence of outliers of a distribution, and it makes intuitive sense that a bandit strategy might depend on some kind of assumption on this quantity. How else to know whether or not to cease exploring an unpromising action? The assumption can also be justified from a mathematical perspective. If the variance of an arm is not assumed known, then calculating confidence intervals requires an estimate of the variance from the data. Let $X, X_1, X_2, \ldots, X_n$ be a sequence of i.i.d. centered random variables with variance $\sigma^2$ and

kurtosis $\kappa$. A reasonable estimate of $\sigma^2$ is

$$\hat{\sigma}^2 = \frac{1}{n} \sum_{t=1}^{n} X_t^2 \,. \tag{2}$$

Clearly this estimator is unbiased and has variance

$$\mathbf{V}[\hat{\sigma}^2] = \frac{\mathbf{E}[X^4] - \mathbf{E}[X^2]^2}{n} = \frac{\sigma^4\,(\kappa - 1)}{n} \,.$$

Therefore, if we are to expect good estimation of $\sigma^2$, then the kurtosis should be finite. Note that if $\sigma^2$ is estimated by (2), then the central limit theorem combined with finite kurtosis is enough for an estimation error of $O(\sigma^2((\kappa - 1)/n)^{1/2})$ *asymptotically*. For bandits, however, finite-time bounds are required, which are not available using (2) without additional moment assumptions (for example, on the moment generating function). An example demonstrating the necessity of the limit in the standard central limit

| Distribution | Parameters | Kurtosis |
|---|---|---|
| Gaussian | $\mu \in \mathbf{R}, \sigma^2 > 0$ | 3 |
| Bernoulli | $\mu \in [0,1]$ | $\frac{1-3\mu(1-\mu)}{\mu(1-\mu)}$ |
| Exponential | $\lambda > 0$ | 9 |
| Laplace | $\mu \in \mathbf{R}, b > 0$ | 9 |
| Uniform | $a < b \in \mathbf{R}$ | $9/5$ |

**Table 2:** Kurtosis

theorem is as follows. Suppose that $X_1, \ldots, X_n$ are Bernoulli with bias $p = 1/n$, then for large $n$ the distribution of the sum is closely approximated by a Poisson distribution with parameter 1, which is very different to a Gaussian. Finite kurtosis alone *is* enough if the classical empirical estimator is replaced by a robust estimator such as the median-of-means estimator [Alon et al., 1996] or Catoni's estimator [Catoni, 2012]. Of course, if the kurtosis were not known, then you could try and estimate it with assumptions on the eighth moment, and so on. Is there any justification to stop here? The main reason is that this seems like a *useful* place to stop. Large classes of distributions have known bounds on their kurtosis (see table) and the independence of scale is a satisfying property.

**Contributions** The main contribution is the new assumption, algorithm, and the proof of Theorem 2 (see §2). The upper bound is also complemented by an asymptotic lower bound (§3) that applies to all strategies with sub-polynomial regret and all bandit problems with bounded kurtosis.

**Additional notation** Let $T_i(t) = \sum_{s=1}^{t} \mathbb{1}\{A_s = i\}$ be the number of times arm $i$ has been played after round $t$. For measures $P, Q$ on the same probability space, $\mathrm{KL}(P, Q)$ is the relative entropy between $P$ and $Q$ and $\chi^2(P, Q)$ is the $\chi^2$ distance. The following lemma is well known.

**Lemma 3.** *Let $X_1, X_2$ be independent random variables with $X_i$ having variance $\sigma_i^2$ and kurtosis $\kappa_i < \infty$ and skewness $\gamma_i = \mathbf{E}[(X_i - \mathbf{E}[X_i])^3/\sigma_i^3]$, then:*

*(a)* $\quad \mathrm{Kurt}[X_1 + X_2] = 3 + \dfrac{\sigma_1^4(\kappa_1 - 3) + \sigma_2^4(\kappa_2 - 3)}{\left(\sigma_1^2 + \sigma_2^2\right)^2}$ *(b)* $\quad \gamma_1 \leq \sqrt{\kappa_1 - 1} \,.$

## 2 Algorithm and upper bound

Like the robust upper confidence bound algorithm by Bubeck et al. [2013], the new algorithm makes use of the robust median-of-means estimator.

**Median-of-means estimator** Let $Y_1, Y_2, \ldots, Y_n$ be a sequence of independent and identically distributed random variables. The median-of-means estimator first partitions the data into $m$ blocks of equal size (up to rounding errors). The empirical mean of each block is then computed and the estimate is the median of the means of each of the blocks. The number of blocks depends on the desired confidence level and should be $O(\log(1/\delta))$. The median-of-means estimator at confidence level $\delta \in (0, 1)$ is denoted by $\widehat{\mathrm{MM}}_\delta((Y_t)_{t=1}^n)$.

**Lemma 4** (Bubeck et al. 2013). *Let $Y_1, Y_2, \ldots, Y_n$ be a sequence of independent and identically distributed random variables with mean $\mu$ and variance $\sigma^2 < \infty$.*

$$\mathbf{P}\left(\left|\widehat{\mathrm{MM}}_\delta\left((Y_t)_{t=1}^n\right) - \mu\right| \geq C_1 \sqrt{\frac{\sigma^2}{n} \log\left(\frac{C_2}{\delta}\right)}\right) \leq \delta \,,$$

*where $C_1 = \sqrt{12 \cdot 16}$ and $C_2 = \exp(1/8)$ are universal constants.*

**Upper confidence bounds** The new algorithm is a generalisation of UCB, but with optimistic estimates of the mean and variance using confidence bounds about the median-of-means estimator. Let $\delta \in (0,1)$ and $Y_1, Y_2, \ldots, Y_t$ be a sequence of independent and identically distributed random variables with mean $\mu$, variance $\sigma^2$ and kurtosis $\kappa < \kappa_\circ$. Furthermore, let

$$\tilde{\mu}((Y_s)_{s=1}^t, \delta) = \sup \left\{ \theta \in \mathbf{R} : \theta \leq \widehat{\mathrm{MM}}_\delta \left((Y_s)_{s=1}^t\right) + C_1 \sqrt{\frac{\tilde{\sigma}_t^2((Y_s)_{s=1}^t, \theta, \delta)}{t} \log \left(\frac{C_2}{\delta}\right)} \right\}.$$

where $\tilde{\sigma}^2((Y_s)_{s=1}^t, \theta, \delta) = \dfrac{\widehat{\mathrm{MM}}_\delta \left(((Y_s - \theta)^2)_{s=1}^t\right)}{\max \left\{ 0, 1 - C_1 \sqrt{\frac{\kappa_\circ - 1}{t} \log \left(\frac{C_2}{\delta}\right)} \right\}}.$

Note that $\tilde{\mu}((Y_s)_{s=1}^t, \delta)$ may be (positive) infinity if $t$ is insufficiently large. The computation of $\tilde{\mu}(\cdot)$ seems non-trivial and is discussed in the summary at the end of the paper where a roughly equivalent and efficiently computable alternative is given. The following two lemmas show that $\tilde{\mu}$ is indeed optimistic with high probability, and also that it concentrates with reasonable speed around the true mean.

**Lemma 5.** $\mathbf{P}\left(\tilde{\mu}((Y_s)_{s=1}^t, \delta) \leq \mu\right) \leq 2\delta$.

*Proof.* By Lemma 4 and the fact that $\mathbf{V}[(Y_s - \mu)^2] = \sigma^4(\kappa - 1) \leq \sigma^4(\kappa_\circ - 1)$ it holds with probability at least $1 - \delta$ that $\tilde{\sigma}^2((Y_s)_{s=1}^t, \mu, \delta) \geq \sigma^2$. Another application of Lemma 4 along with a union bound ensures that with probability at least $1 - 2\delta$,

$$\widehat{\mathrm{MM}}_\delta((Y_s)_{s=1}^t) \leq C_1 \sqrt{\frac{\sigma^2}{t} \log \left(\frac{C_2}{\delta}\right)} \leq C_1 \sqrt{\frac{\tilde{\sigma}_t^2((Y_s)_{s=1}^t, \mu, \delta)}{t} \log \left(\frac{C_2}{\delta}\right)}.$$

Therefore with probability at least $1 - 2\delta$ the true mean $\mu$ is in the set of which $\tilde{\mu}$ is the supremum and in this case $\tilde{\mu}((Y_s)_{s=1}^t, \delta) \geq \mu$ as required. □

**Lemma 6.** *Let $\delta_t$ be monotone decreasing and $\tilde{\mu}_t = \tilde{\mu}((Y_s)_{s=1}^t, \delta_t)$. Then there exists a universal constant $C_3 > 0$ such that for any $\varepsilon > 0$,*

$$\sum_{t=1}^n \mathbf{P}\left(\tilde{\mu}_t \geq \mu + \varepsilon\right) \leq C_3 \max\left\{\kappa_\circ - 1, \frac{\sigma^2}{\varepsilon^2}\right\} \log \left(\frac{C_2}{\delta_n}\right) + 2 \sum_{t=1}^n \delta_t.$$

*Proof.* First, by Lemma 4

$$\sum_{t=1}^n \mathbf{P}\left(\left|\widehat{\mathrm{MM}}_{\delta_t}\left((Y_s)_{s=1}^t\right) - \mu\right| \geq C_1 \sqrt{\frac{\sigma^2}{t} \log \left(\frac{C_2}{\delta_t}\right)}\right) \leq \sum_{t=1}^n \delta_t. \tag{3}$$

Similarly,

$$\sum_{t=1}^n \mathbf{P}\left(\left|\widehat{\mathrm{MM}}_{\delta_t}\left(((Y_s - \mu)^2)_{s=1}^t\right) - \sigma^2\right| \geq C_1 \sigma^2 \sqrt{\frac{\kappa_\circ - 1}{t} \log \left(\frac{C_2}{\delta}\right)}\right) \leq \sum_{t=1}^n \delta_t. \tag{4}$$

Suppose that $t$ is a round where all of the following hold:

(a) $\left|\widehat{\mathrm{MM}}_{\delta_t}\left((Y_s)_{s=1}^t\right) - \mu\right| < C_1 \sqrt{\frac{\sigma^2}{t} \log \left(\frac{C_2}{\delta_t}\right)}$.

(b) $\left|\widehat{\mathrm{MM}}_{\delta_t}\left(((Y_s - \mu)^2)_{s=1}^t\right) - \sigma^2\right| < C_1 \sigma^2 \sqrt{\frac{\kappa_\circ - 1}{t} \log \left(\frac{C_2}{\delta_t}\right)}$.

(c) $t \geq 16 C_1^2 (\kappa_\circ - 1) \log \left(\frac{C_2}{\delta_t}\right)$.

Abbreviating $\tilde{\sigma}_t^2 = \tilde{\sigma}^2((Y_s)_{s=1}^t, \tilde{\mu}_t, \delta_t)$ and $\hat{\mu}_t = \widehat{\mathrm{MM}}_{\delta_t}\left((Y_s)_{s=1}^t\right)$,

$$\tilde{\sigma}_t^2 = \frac{\widehat{\mathrm{MM}}_{\delta_t}\left(((Y_s - \tilde{\mu}_s)^2)_{s=1}^t\right)}{1 - C_1\sqrt{\frac{\kappa_\circ - 1}{t}\log\left(\frac{C_2}{\delta_t}\right)}} \leq 2\widehat{\mathrm{MM}}_{\delta_t}\left(((Y_s - \tilde{\mu}_t)^2)_{s=1}^t\right)$$

$$\leq 4\widehat{\mathrm{MM}}_{\delta_t}\left(((Y_s - \mu)^2)_{s=1}^t\right) + 4(\tilde{\mu}_t - \mu)^2$$

$$\leq 4\widehat{\mathrm{MM}}_{\delta_t}\left(((Y_s - \mu)^2)_{s=1}^t\right) + 8(\tilde{\mu}_t - \hat{\mu}_t)^2 + 8(\hat{\mu}_t - \mu)^2$$

$$< 4\sigma^2 + 4C_1\sigma^2\sqrt{\frac{\kappa_\circ - 1}{t}\log\left(\frac{C_2}{\delta_t}\right)} + \frac{8C_1^2(\sigma^2 + \tilde{\sigma}_t^2)(\kappa_\circ - 1)}{t}\log\left(\frac{C_2}{\delta_t}\right) \leq \frac{11}{2}\sigma^2 + \frac{\tilde{\sigma}_t^2}{2},$$

where the first inequality follows from (c), the second since $(x - y)^2 \leq 2x^2 + 2y^2$ and the fact that

$$\widehat{\mathrm{MM}}_\delta((aY_s + b)_{s=1}^t) = a\widehat{\mathrm{MM}}_\delta((Y_s)_{s=1}^t) + b.$$

The third inequality again uses $(x - y)^2 \leq 2x^2 + 2y^2$, while the last uses the definition of $\tilde{\mu}_t$ and (a,b). Therefore $\tilde{\sigma}_t^2 \leq 11\sigma^2$, which means that if (a,b,c) and additionally

(d) $t \geq \frac{19C_1^2\sigma^2}{\varepsilon^2}\log\left(\frac{1}{\delta_n}\right).$

Then $|\tilde{\mu}_t - \mu| \leq |\tilde{\mu}_t - \hat{\mu}_t| + |\hat{\mu}_t - \mu| < C_1\sqrt{\frac{\tilde{\sigma}_t^2}{t}\log\left(\frac{C_2}{\delta_n}\right)} + C_1\sqrt{\frac{\sigma^2}{t}\log\left(\frac{C_2}{\delta_n}\right)}$

$$\leq C_1\sqrt{\frac{11\sigma^2}{t}\log\left(\frac{C_2}{\delta_n}\right)} + C_1\sqrt{\frac{\sigma^2}{t}\log\left(\frac{C_2}{\delta_n}\right)} \leq \varepsilon.$$

Combining this with (3) and (4) and choosing $C_3 = 19C_1^2$ completes the result. $\qquad\square$

**Algorithm and Proof of Theorem 2**    Let $\delta_t = 1/(t^2\log(1+t))$ and $\tilde{\mu}_i(t) = \tilde{\mu}((X_s)_{s\in[t], A_s=i}, \delta_t)$. In each round the algorithm chooses $A_t = \arg\max_{i\in[k]}\tilde{\mu}_i(t-1)$, where ties are broken arbitrarily.

*Proof of Theorem 2.* Assume without loss of generality that $\mu_1 = \mu^*$. Then suboptimal arm $i$ is only played in round $t$ if either $\tilde{\mu}_1(t-1) \leq \mu_1$ or $\tilde{\mu}_i(t-1) \geq \mu_1$. Therefore

$$\mathbf{E}[T_i(n)] \leq \sum_{t=1}^n \mathbf{P}\left(\tilde{\mu}_1(t-1) \leq \mu_1\right) + \sum_{t=1}^n \mathbf{P}\left(\tilde{\mu}_i(t-1) \geq \mu_1 \text{ and } A_t = i\right) \qquad (5)$$

The two sums are bounded using Lemmas 5 and 6 respectively:

$$\sum_{t=1}^n \mathbf{P}\left(\tilde{\mu}_1(t-1) \leq \mu_1\right) \leq \sum_{t=1}^n \sum_{u=1}^t \mathbf{P}\left(\tilde{\mu}_1(t-1) \leq \mu_1 \text{ and } T_1(t-1) = u\right)$$

$$\leq 2\sum_{t=1}^n \sum_{u=1}^t \delta_t = 2\sum_{t=1}^n t\delta_t = o(\log(n)). \qquad \text{(By Lem. 5)}$$

$$\sum_{t=1}^n \mathbf{P}\left(\tilde{\mu}_i(t-1) \geq \mu_1 \text{ and } A_t = i\right) \leq \sum_{t=1}^n \mathbf{P}\left(\tilde{\mu}_i(t-1) - \mu_i \geq \Delta_i\right)$$

$$\leq C_3\max\left\{\kappa_\circ - 1, \frac{\sigma_i^2}{\Delta_i^2}\right\}\log\left(\frac{C_2}{\delta_n}\right) + 2\sum_{t=1}^n \delta_t = o(\log(n)). \qquad \text{(By Lem. 6)}$$

And the result follows by substituting the above bounds into Eq. (5) and then into the regret decomposition $\mathcal{R}_n = \sum_{i=1}^k \Delta_i\mathbf{E}[T_i(n)]$. $\qquad\square$

# 3 Lower bound

Let $\mathcal{H}_{\kappa_\circ} = \{\nu : \nu \text{ has kurtosis less than } \kappa_\circ\}$ be the class of all distributions with kurtosis bounded by $\kappa_\circ$. Following the nomenclature of Lai and Robbins [1985], a bandit strategy is called *consistent* over $\mathcal{H}$ if $\mathcal{R}_n = o(n^p)$ for all $p \in (0,1)$ and bandits $(\nu_i)_i$ with $\nu_i \in \mathcal{H}_{\kappa_\circ}$ for all $i$. The next theorem shows that the upper bound derived in the previous section is nearly tight up to constant factors. Let $\mathcal{H}$ be a family of distributions and let $(\nu_i)_i$ be a bandit with $\nu_i \in \mathcal{H}$ for all $i$. Burnetas and Katehakis [1996] showed that for any consistent strategy,

$$\text{for all } i \in [k]: \quad \liminf_{n \to \infty} \frac{\mathbf{E}[T_i(n)]}{\log(n)} \geq \left(\inf\left\{\mathrm{KL}(\nu_i, \nu_i') : \nu_i' \in \mathcal{H} \text{ and } \mathbf{E}_{X \sim \nu_i'}[X] > \mu^*\right\}\right)^{-1}. \quad (6)$$

In parameterised families of distributions, the optimisation problem can often be evaluated analytically (eg., Bernoulli, Gaussian with known variance, Gaussian with unknown variance, Exponential). For non-parametric families the calculation is much more challenging. The following theorem takes the first steps towards understanding this problem for the class of distributions $\mathcal{H}_{\kappa_\circ}$ for $\kappa_\circ \geq 7/2$.

**Theorem 7.** *Let $\kappa_\circ \geq 7/2$ and $\Delta > 0$ and $\nu \in \mathcal{H}_{\kappa_\circ}$ with mean $\mu$, variance $\sigma^2 > 0$ and kurtosis $\kappa$. Then for appropriately chosen universal constant $C, C' > 0$,*

$$\inf\left\{\mathrm{KL}(\nu, \nu') : \nu' \in \mathcal{H}_\kappa \text{ and } \mathbf{E}_{X \sim \nu'}[X] > \mu + \Delta\right\} \leq \frac{7}{5}\min\left\{\frac{1}{\kappa_\circ}, \frac{\Delta}{\sigma}\right\}.$$

*If additionally it holds that $\kappa + C'\Delta\kappa^{1/2}(\kappa + 1) \leq \kappa_\circ$, then*

$$\inf\left\{\mathrm{KL}(\nu, \nu') : \nu' \in \mathcal{H}_\kappa \text{ and } \mathbf{E}_{X \sim \nu'}[X] > \mu + \Delta\right\} \leq C\frac{\Delta^2}{\sigma^2}$$

Therefore provided that $\nu \in \mathcal{H}_{\kappa_\circ}$ is not too close to the boundary of $\mathcal{H}_{\kappa_\circ}$ in the sense that its kurtosis is not too close to $\kappa_\circ$, then the lower bound derived from Theorem 7 and Eq. (6) matches the upper bound up to constant factors. This condition is probably necessary because distributions like the Bernoulli with kurtosis close to $\kappa_\circ$ have barely any wiggle room to increase the mean without also increasing the kurtosis.

*Proof of Theorem 7.* Let $\Delta_\varepsilon = \Delta + \varepsilon$ for small $\varepsilon > 0$. Assume without loss of generality that $\nu$ is centered and has variance $\sigma^2 = 1$, which can always be achieved by shifting and scaling (neither effects the kurtosis or the relative entropy). The first part of the claim is established by considering the perturbed distribution obtained by adding a Bernoulli 'outlier'. Let $X$ be a random variable sampled from $\nu$ and $B$ be a Bernoulli with parameter $p = \min\{\Delta_\varepsilon, 1/\kappa_\circ\}$. Let $Z = X + Y$ where $Y = \Delta_\varepsilon B/p$. Then $\mathbf{E}[Z] = \Delta_\varepsilon > \Delta$ and

$$\mathrm{Kurt}[Z] = 3 + \frac{\kappa - 3 + \mathbf{V}[Y]^2(\mathrm{Kurt}[Y] - 3)}{(1 + \mathbf{V}[Y])^2} = 3 + \frac{\kappa - 3 + \left(\frac{(1-p)\Delta_\varepsilon^2}{p}\right)^2 \frac{1 - 6p(1-p)}{p(1-p)}}{\left(1 + \frac{(1-p)\Delta_\varepsilon^2}{p}\right)^2}$$

$$\leq 3 + \frac{\kappa_\circ - 3 + \left(\frac{(1-p)\Delta_\varepsilon^2}{p}\right)^2 \frac{1 - 6p(1-p)}{p(1-p)}}{\left(1 + \frac{(1-p)\Delta_\varepsilon^2}{p}\right)^2} \leq \kappa_\circ,$$

where the first inequality used Lemma 3 and the final inequality follows from simple case-based analysis, calculus and the assumption that $\kappa_\circ \geq 7/2$ (see Lemma 9 in the appendix). Let $\nu' = \mathcal{L}(Y)$ be the law of $Y$. Then

$$\mathrm{KL}(\nu, \nu') = \int_{\mathbf{R}} \log\frac{d\nu}{d\nu'}d\nu \leq \int_{\mathbf{R}} \log\frac{1}{1-p}d\nu = \log\frac{1}{1-p} \leq \frac{p}{1-p} \leq \frac{7}{5}\min\left\{\Delta_\varepsilon, \frac{1}{\kappa_\circ}\right\}.$$

Taking the limit as $\varepsilon$ tends to $0$ completes the proof of the first part of the theorem. Moving onto the second claim and using $C$ for a universal positive constant that changes from equation to equation. Let $a > 0$ be a constant to be chosen later and $A = \{x : |x| \leq \sqrt{a\kappa}\}$ and $\bar{A} = \mathbf{R} - A$. Define

alternative measure $\nu'(E) = \int_E (1+g)d\nu$ where $g(x) = (\alpha + \beta x)\mathbb{1}\{x \in A\}$ for some constants $\alpha$ and $\beta$ chosen so that

$$\int_{\mathbf{R}} g(x)d\nu(x) = \alpha \int_A d\nu(x) + \beta \int_A x d\nu(x) = 0\,.$$

$$\int_{\mathbf{R}} g(x)x d\nu(x) = \alpha \int_A x d\nu(x) + \beta \int_A x^2 d\nu(x) = \Delta_\varepsilon\,.$$

Solving for $\alpha$ and $\beta$ shows that

$$\beta = \frac{\Delta_\varepsilon}{\int_A x^2 d\nu(x) - \frac{\left(\int_A x d\nu(x)\right)^2}{\nu(A)}} \quad \text{and} \quad \alpha = -\frac{\Delta_\varepsilon \int_A x d\nu(x)}{\nu(A)\int_A x^2 d\nu(x) - \left(\int_A x d\nu(x)\right)^2}\,.$$

This implies that $\int_{\mathbf{R}} d\nu'(x) = 1$ and $\int_{\mathbf{R}} x d\nu'(x) = \Delta_\varepsilon > \Delta$. It remains to show that $\nu'$ is a probability measure with kurtosis bounded by $\kappa_\circ$. That $\nu'$ is a probability measure will follow from the positivity of $1 - g(\cdot)$. The first step is to control each of the terms appearing in the definitions of $\alpha$ and $\beta$. By Cauchy-Schwarz and Chebyshev's inequalities, $\nu(\bar{A}) = \nu(x^2 \geq a\kappa) \leq 1/(\kappa a^2)$ and

$$\int_A x^2 d\nu(x) = 1 - \int_{\bar{A}} x^2 d\nu(x) \geq 1 - \sqrt{\kappa \nu(\bar{A})} \geq 1 - \frac{1}{a}\,.$$

Similarly, since $\nu$ is centered,

$$\left|\int_A x d\nu(x)\right| = \left|\int_{\bar{A}} x d\nu(x)\right| \leq \sqrt{\sigma^2 \nu(\bar{A})} \leq \frac{1}{a\sqrt{\kappa}}\,.$$

Therefore by choosing $a = 2$ and using the fact that the kurtosis is always larger than 1,

$$|\alpha| = \Delta_\varepsilon \left|\frac{\int_A x d\nu(x)}{\nu(A)\int_A x^2 d\nu(x) - \left(\int_A x d\nu(x)\right)^2}\right| \leq \frac{\Delta_\varepsilon/\sqrt{\kappa}}{a\left(\left(1 - \frac{1}{\kappa a^2}\right)\left(1 - \frac{1}{a}\right) - \frac{1}{a^2\kappa}\right)} \leq \frac{4\Delta_\varepsilon}{\sqrt{\kappa}}$$

$$|\beta| = \frac{\Delta_\varepsilon}{\int_A x^2 d\nu(x) - \frac{\left(\int_A x d\nu(x)\right)^2}{\nu(A)}} \leq \frac{\Delta_\varepsilon}{1 - \frac{1}{a} - \frac{1}{\kappa a^2\left(1 - \frac{1}{a^2\kappa}\right)}} \leq 6\Delta_\varepsilon\,.$$

Now $g(x)$ is a linear function supported on compact set $A$, so

$$\max_{x \in \mathbf{R}} |g(x)| = \max\left\{|g(\sqrt{a\kappa})|, |g(-\sqrt{a\kappa})|\right\} \leq |\alpha| + \sqrt{a\kappa}|\beta| \leq \frac{4\Delta_\varepsilon}{\sqrt{\kappa}} + 6\Delta_\varepsilon\sqrt{2\kappa} \leq \frac{1}{2}\,,$$

where the last inequality follows by assuming that $\Delta_\varepsilon \leq \sqrt{\kappa}/(4(2 + 3\sqrt{2}\kappa)) = O(\kappa^{-1/2})$, which is reasonable without loss of generality, since if $\Delta_\varepsilon$ is larger than this quantity, then we would prefer the bound that depends on $\kappa_\circ$ derived in the first part of the proof. The relative entropy between $\nu$ and $\nu'$ is bounded by

$$\mathrm{KL}(\nu, \nu') \leq \chi^2(\nu, \nu') = \int_{\mathbf{R}} \left(\frac{d\nu(x)}{d\nu'(x)} - 1\right)^2 d\nu'(x) = \int_A \frac{g(x)^2}{1 + g(x)} d\nu(x)$$

$$\leq 2 \int_A g(x)^2 d\nu(x) \leq 4 \int_A \alpha^2 d\nu(x) + 4 \int_A \beta^2 x^2 d\nu(x) \leq 4\alpha^2 + 4\beta^2$$

$$\leq \frac{4 \cdot 16\Delta_\varepsilon^2}{\kappa} + 4 \cdot 36\Delta_\varepsilon^2 \leq C\Delta_\varepsilon^2\,.$$

In order to bound the kurtosis we need to evaluate the moments:

$$\int_{\mathbf{R}} x^2 d\nu' = \int_{\mathbf{R}} x^2 d\nu + \int_A g(x)x^2 d\nu = 1 + \alpha \int_A x^2 d\nu(x) + \beta \int_A x^3 d\nu(x) \leq 1 + C\Delta_\varepsilon\sqrt{\kappa}\,.$$

$$\int_{\mathbf{R}} x^2 d\nu' = \int_{\mathbf{R}} x^2 d\nu + \int_A g(x)x^2 d\nu \geq 1 - C\Delta_\varepsilon\sqrt{\kappa}\,.$$

$$\int_{\mathbf{R}} x^4 d\nu' = \int_{\mathbf{R}} x^4 d\nu + \int_A g(x)x^4 d\nu = \kappa + \alpha \int_A x^4 d\nu(x) + \beta \int_A x^5 d\nu(x) \leq \kappa\left(1 + C\Delta_\varepsilon\sqrt{\kappa}\right)\,.$$

$$\left|\int_{\mathbf{R}} x^3 d\nu'(x)\right| \leq \sqrt{\int_{\mathbf{R}} x^2 d\nu'(x) \int_{\mathbf{R}} x^4 d\nu'(x)} \leq \sqrt{C\kappa}\,.$$

Therefore if $\kappa'$ is the kurtosis of $\nu'$, then

$$\kappa' = \frac{\int_{\mathbf{R}}(x - \Delta_\varepsilon)^4 d\nu'(x)}{\left(\int_{\mathbf{R}} x^2 d\nu'(x) - \Delta_\varepsilon^2\right)^2} = \frac{\int_{\mathbf{R}} x^4 d\nu'(x) - 3\Delta_\varepsilon^4 + 6\Delta_\varepsilon^2 \int_{\mathbf{R}} x^2 d\nu'(x) - 4\Delta_\varepsilon \int_{\mathbf{R}} x^3 d\nu'(x)}{\left(1 - \Delta_\varepsilon^2 + \alpha \int_A x^2 d\nu(x) + \beta \int_A x^3 d\nu(x)\right)^2}$$

Therefore

$$
\begin{aligned}
\kappa' &= \frac{\int_{\mathbf{R}} x^4 d\nu'(x) - 3\Delta_\varepsilon^4 + 6\Delta_\varepsilon^2 \int_{\mathbf{R}} x^2 d\nu'(x) - 4\Delta_\varepsilon \int_{\mathbf{R}} x^3 d\nu'(x)}{\left(\int_{\mathbf{R}} x^2 d\nu'(x) - \Delta_\varepsilon^2\right)^2} \\
&\leq \frac{\kappa\left(1 + C\Delta_\varepsilon \kappa^{1/2}\right) + 6\Delta_\varepsilon^2(1 + C\Delta_\varepsilon \kappa^{1/2}) + C\Delta_\varepsilon \kappa^{1/2}}{\left(1 - C\Delta_\varepsilon \kappa^{1/2} - \Delta_\varepsilon^2\right)^2} \\
&\leq \frac{\kappa + C\Delta_\varepsilon \kappa^{1/2}(\kappa + 1)}{1 - C\Delta_\varepsilon \kappa^{1/2}} \leq \kappa + C\Delta_\varepsilon \kappa^{1/2}(\kappa + 1) \, .
\end{aligned}
$$

Therefore $\kappa' \leq \kappa_\circ$ provided $\Delta_\varepsilon$ is sufficiently small, which after taking the limit as $\varepsilon \to 0$ completes the proof. □

## 4  Summary

The assumption of finite kurtosis generalises the parametric Gaussian assumption to a comparable non-parametric setup with a similar basic structure. Of course there are several open questions.

**Optimal constants**   The leading constants in the main results (Theorem 2 and Theorem 7) are certainly quite loose. Deriving the optimal form of the regret is an interesting challenge, with both lower and upper bounds appearing quite non-trivial. It may be necessary to resort to an implicit analysis showing that (6) is (or is not) achievable when $\mathcal{H}$ is the class of distributions with kurtosis bounded by some $\kappa_\circ$. Even then, constructing an efficient algorithm would remain a challenge. Certainly what has been presented here is quite far from optimal. At the very least the median-of-means estimator needs to be replaced, or the analysis improved. An excellent candidate is Catoni's estimator [Catoni, 2012], which is slightly more complicated than the median-of-means, but also comes with smaller constants and could be plugged into the algorithm with very little effort. An alternative approach is to use the theory of self-normalised processes [Peña et al., 2008], but even this seems to lead to suboptimal constants. For the lower bound, there appears to be almost no work on the explicit form of the lower bounds presented by Burnetas and Katehakis [1996] in interesting non-parametric classes beyond rewards with bounded or semi-bounded support [Honda and Takemura, 2010, 2015].

**Absorbing other improvements**   There has recently been a range of improvements to the confidence level for the classical upper confidence bound algorithms that shave logarithmic terms from the worst-case regret or improve the lower-order terms in the finite-time bounds [Audibert and Bubeck, 2009, Lattimore, 2015]. Many of these enhancements can be incorporated into the algorithm presented here, which may lead to practical and theoretical improvements.

**Computation complexity**   The main challenge is the computation of the index, which as written seems challenging. The easiest solution is to change the algorithm slightly by estimating

$$\hat{\mu}_i(t) = \widehat{\mathrm{MM}}_{\delta_t}((X_s)_{s\in[t], A_s=i}) \qquad \hat{\sigma}_i^2(t) = \widehat{\mathrm{MM}}_{\delta_t}((X_s^2)_{s\in[t], A_s=i}) - \hat{\mu}_i(t)^2 \, .$$

Then an upper confidence bound on $\hat{\mu}_i(t)$ is easily derived from Lemma 4 and the rest of the analysis goes through in about the same way. Naively the computational complexity of the above is $\Omega(t)$ in round $t$, which would lead to a running time over $n$ rounds of $\Omega(n^2)$. Provided the number of buckets used between rounds $t$ and $t+1$ is the same, then the median-of-means estimator can be updated incrementally in $O(B_t)$ time, where $B_t$ is the number of buckets. Now $B_t = O(\log(1/\delta_t)) = O(\log(t))$ so there are at most $O(\log(n))$ changes over $n$ rounds. Therefore the total computation is $O(nk + n\log(n))$.

**Comparison to Bernoulli** Table 2 shows that the kurtosis for a Bernoulli random variable with mean $\mu$ is $\kappa = O(1/(\mu(1-\mu)))$, which is obviously not bounded as $\mu$ tends towards the boundaries. The optimal asymptotic regret for the Bernoulli case is $\lim_{n\to\infty} \mathcal{R}_n/\log(n) = \sum_{i:\Delta_i>0} \Delta_i/d(\mu_i, \mu^*)$. The interesting differences occur near the boundary of the parameter space. Suppose that $\mu_i \approx 0$ for some arm $i$ and $\mu^* > 0$ is close to zero. An easy calculation shows that $d(\mu_i, \mu^*) \approx \log(1/(1-\Delta_i)) \approx \Delta_i$. Therefore

$$\liminf_{n\to\infty} \frac{\mathbf{E}[T_i(n)]}{\log(n)} \approx \frac{1}{\log(1/(1-\Delta_i))} \approx \frac{1}{\Delta_i}\,.$$

Here we see an algorithm is enjoying logarithmic regret on a class with infinite kurtosis! But this is a special case and is not possible in general. The reason is that the structure of the hypothesis class allows strategies to (essentially) estimate the kurtosis with reasonable accuracy and anticipate outliers more/less depending on the data observed so far. Another way of saying it is that when the kurtosis is actually small, the algorithms can learn this fact by examining the empirical mean.

## A  Technical calculations

This section completes some of the calculations required in the proof of Theorem 7.

**Lemma 8.** *Let $\kappa_\circ \geq 7/2$ and $f(x) = 3 + (\kappa_\circ - 3 + x)/(1+x)^2$. Then $f(x) \leq \kappa_\circ$ for all $x \geq 0$.*

*Proof.* Clearly $f(0) = \kappa_\circ$ and for $\kappa_\circ \geq 7/2$ and $x \geq 0$,

$$f'(x) = \frac{1}{(1+x)^2}\left(1 - \frac{2(\kappa_\circ - 3 + x)}{1+x}\right) \leq 0\,.$$

Therefore $f(x) = \kappa_\circ + \int_0^x f'(y)dy \leq \kappa_\circ$. $\qquad\square$

**Lemma 9.** *If $\kappa_\circ \geq 7/2$ and $p = \min\{\Delta, 1/\kappa_\circ\}$, then*

$$3 + \frac{\kappa_\circ - 3 + \left(\frac{(1-p)\Delta^2}{p}\right)^2 \frac{1-6p(1-p)}{p(1-p)}}{\left(1 + \frac{(1-p)\Delta^2}{p}\right)^2} \leq \kappa_\circ\,.$$

*Proof.* Suppose that $p = \Delta$. Then since $\kappa_\circ \geq 7/2 \geq 1$, $p \leq 1$. Therefore

$$3 + \frac{\kappa_\circ - 3 + \left(\frac{(1-p)\Delta^2}{p}\right)^2 \frac{1-6p(1-p)}{p(1-p)}}{\left(1 + \frac{(1-p)\Delta^2}{p}\right)^2} = 3 + \frac{\kappa_\circ - 3 + \Delta(1-\Delta)(1 - 6\Delta(1-\Delta))}{(1 + \Delta(1-\Delta))^2}$$

$$\leq 3 + \frac{\kappa_\circ - 3 + \Delta(1-\Delta)}{(1 + \Delta(1-\Delta))^2}$$

$$\leq \kappa_\circ\,,$$

where the last inequality follows from Lemma 8. Now suppose that $p = 1/\kappa_\circ$. Then

$$3 + \frac{\kappa_\circ - 3 + \left(\frac{(1-p)\Delta^2}{p}\right)^2 \frac{1-6p(1-p)}{p(1-p)}}{\left(1 + \frac{(1-p)\Delta^2}{p}\right)^2} \leq 3 + \frac{\kappa_\circ - 3 + \left(\frac{(1-p)\Delta^2}{p}\right)^2 \frac{1-6p(1-p)}{p(1-p)}}{1 + \left(\frac{(1-p)\Delta^2}{p}\right)^2}$$

$$\leq \max\left\{\kappa_\circ, \frac{\kappa_\circ}{1 - \frac{1}{\kappa_\circ}} - 3\right\}$$

$$\leq \kappa_\circ\,,$$

where the first inequality follows since $(a+b)^2 \geq a^2 + b^2$ for $a, b \geq 0$. The second since the average is less than the maximum. The third since $\kappa_\circ \geq 7/2 > 4/3$. $\qquad\square$

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
