[Reviews · NeurIPS 2017]

Reviewer 1



The paper provides both upper (Theorem 2) and lower (Theorem 7) bounds for the regret of a finite-armed stochastic bandit model and a UCB type algorithm that is based on confidence bounds for the median-of-means estimator, using work of Bubeck et al 2013. The analysis is done under a non-parametric setup, where the learner need to only assume the existence of a bound on the kurtosis of the noise. I read the paper and I found it to be well written and the topic to be very interesting. The author(s) did a wonderful job describing the importance of the work, and to delineate the proposed solution.The manuscript is in great shape. I can therefore recommend the acceptance of the paper subject to a minor revision: The reference: Wesley Cowan and Michael N Katehakis. Normal bandits of unknown means and variances: Asymp- totic optimality, finite horizon regret bounds, and a solution to an open problem. arXiv preprint arXiv:1504.05823, 2015a. Should be replaced by the most recent: Wesley Cowan, Junya Honda, Michael N. Katehakis Normal bandits of unknown means and variances: Asymptotic optimality, finite horizon regret bounds, and a solution to an open problem. arXiv preprint arXiv:1504.05823v2, 2015a. 2. In table 2 the authors should include the result for arbitrary discrete distributions of Burnetas and Katehakis (1996), with a complicated constant. (Section 4.3 of that paper).

Reviewer 2



Rebuttal -------- I read the author feedback. There was no discussion, but due to the consensus, the paper should be accepted. Summary ------- The article considers a variant of the Upper Confidence Bound (UCB) algorithm for multi-armed bandit that is using the notion of kurtosis as well as median of means estimates for the mean and the variance of the considered distributions. Detailed comments ----------------- 1) Table 1: I think the work by Burnetas and Katehakis regarding bandit lower bounds for general distributions deserves to be in this table. 2) It is nice to provide Theorem 7 and make explicit the lower bound in (special cases of) this setting. 3) The algorithm that is derived looks very natural in view of the discussion regarding the kurtosis. One may observe that a bound kappa_0 on kappa is currently required by the algorithm, and I suggest you add some paragraph regarding this point: Indeed on the one hand, a (naive) criticism is that people may not know a value kappa_0 in practice and thus one would now like to know what can be achieved when the kurtosis is unknown and we want to estimate it; This would then call for another assumption such as on higher order moments, and one may then continue asking similar questions for the novel assumption, and so on and so forth. On the other hand, the crucial fact about kurtosis is that unlike mean or variance, for many classes of distributions all distributions of a certain type share the same kurtosis. Thus this is rather a "class" dependent quantity than a distribution dependent quantity, and using the knowledge of kappa_0 \geq \kappa can be considered to be a weak assumption. I think this should be emphasized more. 4) Proof of Lemma 5: The proof is a little compact. I can see the missing steps; but for the sake of clarity, it may be nice to reformulate/expand slightly. (Applying Lemma 4 twice, first to the random variables Y_s, then to (Y_s-mu)^2, etc). 5) Proof of Lemma 6: end of page 4, first inequality: from (c), you can improve the factor 2 to 4/3 (1/ (1-1/4) instead of 1/(1-1/2)) Line 112 "while the last uses the definition..." and Lemma 4 as well? With the 2/3 improvement, you should get \tilde \sigma_t^2 \leq 11/3 \sigma^2 + \tilde \sigma_t^2/3. 6) Without section 3, the paper would have been a little weak, since the proof techniques are extremely straightforward once we have the estimation errors of the median of means, which are known. Thus this section is welcomed. 7) Proof of Theorem 7: "Let Delta_\epsilon = \Delta_\epsilon + \epsilon" I guess it should be "Let Delta_\epsilon = \Delta + \epsilon". 8) There are some missing words on page 7. 9) Section 4: i)Optimal constants. For the seek improvement, one should mimic the KL optimization, at least implicitly. I am not sure this can be achieved that easily. Self-normalized processes would play a role in reducing the magnitude of the confidence bounds, I am not sure it will help beyond that. ii) TS: I don't see any simple way to define a "kurtosis-based" prior/posterior. Can you elaborate on this point? Otherwise this seems like an empty paragraph and I suggest you remove it. Decision: -------- Overall, this is an article with a rather simple (a direct application of median of means confidence bounds for unknown mean and variance) but elegant idea, and the paper is nicely executed and very clear. I give a weak accept, not a clear accept only because most of the material is already known.

Reviewer 3



The paper provides a UCB-type algorithm for bandits with reward distribution with bounded kurtosis. The strategy of the paper is to use robust median of means estimators to get finite time convergence. From lines 55 to 68, it is suggested that the classical empirical mean estimators won't do the job. I think it would be very helpful if the author gives a concrete example of an empirical estimator failing here. A major concern about the paper is that it has typos in its definitions and proofs. It can sometimes be frustrating to follow the arguments. As an example, I believe definition of T_i on line 72, and eqn (5) after line 118 to have this kind of issues. Moreover, after line 119, shouldn't u start from 0 and go to t? Also line 142 doesn't make sense! I suggest adding more explanations to the proof of Theorem 2 specially the few lines after 119. I wasn't able to follow some of the conclusions.